# Hybrid Grid Pattern Star Identification Algorithm Based on Multi-Calibration Star Verification

**DOI:** 10.3390/s24051661

**Published:** 2024-03-04

**Authors:** Chao Shen, Caiwen Ma, Wei Gao, Yuanbo Wang

**Affiliations:** Xi’an Institute of Optics and Precision Mechanics of Chinese Academy of Sciences, Xi’an 710119, China; shenchao@opt.ac.cn (C.S.); gaowei@opt.ac.cn (W.G.); wangyuanbo@opt.ac.cn (Y.W.)

**Keywords:** star identification, hybrid grid pattern, reference star map verification, multi-calibration star verification

## Abstract

In order to solve the star identification problem in the lost space mode for scientific cameras with small fields of view and higher instruction magnitudes, this paper proposes a star identification algorithm based on a hybrid grid pattern. The application of a hybrid pattern generated by multi-calibration stars in the initial matching enables the position distribution features of neighboring stars around the main star to be more comprehensively described and avoids the interference of position noise and magnitude noise as much as possible. Moreover, calibration star filtering is adopted to eliminate incorrect candidates and pick the true matched navigation star from candidate stars in the initial match. Then, the reference star image is utilized to efficiently verify and determine the final identification results of the algorithm via the nearest principle. The performance of the proposed algorithm in simulation experiments shows that, when the position noise is 2 pixels, the identification rate of the algorithm is 96.43%, which is higher than that of the optimized grid algorithm by 2.21% and the grid algorithm by 4.05%; when the magnitude noise is 0.3 mag, the star identification rate of the algorithm is 96.45%, which is superior to the optimized grid algorithm by 2.03% and to the grid algorithm by 3.82%. In addition, in the actual star image test, star magnitude values of ≤12 mag can be successfully identified using the proposed algorithm.

## 1. Introduction

Star sensors [1] play a crucial role in attitude control systems and are extensively utilized in deep space exploration missions due to their desirable characteristics including no drift, high reliability, and arcsecond-level accuracy in attitude measurement, which surpasses that of other attitude determination devices, such as solar sensors, geomagnetic sensors, and gyroscopes. However, with the emergence of the trend toward satellite miniaturization, the factors of device weight, size, and cost have become increasingly significant. Traditional star sensors with 12 deg fields of view (FOV) are typically characterized by their expensive costs, substantial size, and increased power consumption.

Since most micro-spacecrafts usually carry a science camera to acquire visual images, it can be utilized as an alternative solution to specialized star sensors to obtain star images and determine attitude. Reducing the number of instruments decreases spacecraft weight and complexity, thus resulting in cost savings for the mission. In addition, the typical FOV of the science camera is small (2 deg or less). It can be anticipated that the final attitude estimate obtained from a science camera image will exhibit higher accuracy in pitch and yaw compared to that achieved with a standard star sensor. But the limited FOV of the science camera also introduces some challenges: expanding the on-board star catalog to include dimmer stars will necessitate additional on-board memory and make it difficult to identify stars due to the larger number of possibilities that need to be distinguished. 

Since the first star identification algorithm was proposed in 1979, many star identification algorithms have been designed to estimate attitude data without any prior attitude data, and several algorithms [2,3,4,5] have been verified in engineering practice for star sensing. The existing star identification algorithms are generally divided into three categories: subgraph isomorphism algorithms [6,7], artificial intelligence algorithms [8], and pattern recognition algorithms [9].

A subgraph isomorphism algorithm treats the star recognition problem as a specialized case of the subgraph isomorphism problem. In this type of star identification algorithm, star points within the field of view (FOV) are considered to be the vertices of a graph, and the distance (angular distance) between the star points is used as the edges. This approach constructs geometric shapes and searches for subgraphs in a database that matches the undirected shapes obtained using the star image. Typically, the distance between star pairs, triangles, or polygons is used to build a navigation database. Common subgraph isomorphism algorithms include the triangle algorithm, pyramid algorithm [10], matching group algorithm, geometric voting method [4], simplest subgraph method [11], vector matching method [12], and others. The subgraph isomorphism algorithm exhibits a certain level of robustness against positional and magnitude noise. It requires a large database capacity to store subgraph features. However, it has a slow searching capacity, low dimensionality of feature subgraphs, unreliable matching results, and limited robustness in the subgraph construction process.

Due to the rapid advancements in artificial intelligence, numerous algorithms have emerged in recent years for star identification. These algorithms primarily rely on neural networks or biological intelligence algorithms to recognize navigation star feature vectors. Neural networks excel in their ability to learn from data, allowing them to deeply analyze patterns during training and effectively recognize distorted observation patterns. They also exhibit low complexity in online searching and matching tasks. Typical algorithms in this category include the improved grid pattern star recognition algorithm based on backpropagation (BP) networks [13], the mixed feature star pattern recognition algorithm based on one-dimensional convolutional networks [14], and the radial star pattern recognition algorithm based on multi-layer representation learning networks (RPNet) [15]. However, the aforementioned algorithms still require the manual design of star pattern features, and the neural network only handles the online feature pattern-matching task. In contrast to the neural network algorithms mentioned above, Liu Lei [8] connects the original star map according to certain rules before the identification step, forming a colored spider web graph that provides edge information for network learning. However, this algorithm has limited application value due to the high cost of collecting the training dataset and its overall complexity. Additionally, star identification involves optimization and combination problems during the feature matching process. Therefore, biological intelligence optimization algorithms such as ant colony algorithms and genetic algorithms can be utilized to address these challenges. However, online calculations involving biological intelligence optimization algorithms are computationally intensive and often involve random variables. This can pose significant challenges in practical applications, as the algorithms may require extensive computational resources and may not always provide consistent results due to the presence of random variables.

The pattern recognition algorithm in star identification involves constructing a unique pattern or marker for each navigation star, typically based on its neighboring star distribution. The most similar pattern in the navigation star catalog to that observed is considered to be a matching star. The grid algorithm [16], proposed by Padgett in 1997, was one of the earliest pattern-based star pattern recognition algorithms. This offers faster processing speed, higher recognition rates, and greater robustness to noise compared to other algorithms. However, the grid algorithm has a drawback in that it requires the selection of neighboring stars to construct the grid pattern, and the probability of correctly selecting neighboring stars is low, which results in incorrect matching or identification failures. Scholars such as Na [17], Clouse [18], and Lee [19] have proposed improvement strategies, but the fundamental flaw in the algorithm has not been fully addressed. The Italian scholar Bittant introduced the pole algorithm [20], which is based on radial feature patterns and overcomes the sensitivity of the grid algorithm to calibration stars. However, it only partially describes the distribution characteristics of neighboring stars, leading to decreased performance when subjected to significant positional noise. Subsequent researchers, including Zhang Guangjun [21], Wei Xin [22], and Chen Shoushun [23], have proposed circular distribution description operators and optimization matching strategies to more comprehensively capture the positional distribution information of neighboring stars. Wei [24] introduced a circumferential distribution description operator and an optimization matching strategy to enhance the robustness of radial patterns. Furthermore, new approaches have emerged to describe the distribution characteristics of neighboring stars using log-polar transformations [25], one-dimensional vector patterns, wheel encoding features, integer labels, composite main-star pole patterns, and triangular feature encoding patterns. However, the construction of feature patterns in these algorithms is still sensitive to interference from incorrect calibration stars, and the pattern-matching process remains complex.

The aforementioned star identification algorithms are mainly used for star sensors whose FOV is greater than 12° and whose limiting magnitude is below 6.5 mag, and these algorithms cannot be directly applied to the star identification problem of narrow-field-of-view scientific cameras (less than or equal to 2°) with higher limiting magnitudes (≥12 mag) across the entire celestial sphere. In the case of a camera with a limiting magnitude of 12, it is capable of detecting approximately $2.8\times 10^{6}$ stars, which is approximately 310 times greater than the number of stars detectable by a 6.5-magnitude star sensor. Based on the above analysis, it can be concluded that due to the presence of a large number of faint stars in the field of view, algorithms need to perform star matching among a vast number of navigation stars across the celestial sphere. In comparison to subgraph isomorphism algorithms, pattern recognition algorithms have higher feature dimensions constructed for navigation stars. Therefore, this technological solution has the potential to achieve feature distinction among navigation stars and is more suitable for star pattern recognition tasks in narrow-field-of-view scientific cameras. Therefore, the core of the problem lies in defining a pattern that accurately and effectively describes the distribution of neighboring stars, while also possessing strong robustness against positional noise and magnitude noise.

This dissertation proposes a hybrid grid pattern algorithm based on multi-calibration star verification for the recognition and detection of dim stars in a specific sub-sky area. The algorithm utilizes a mixed grid pattern composed of a grid pattern and we use the nearest-neighbor star angle vector as the matching vector. Additionally, multiple calibration stars are introduced during the pattern construction process to enhance robustness. Subsequently, secondary verification is conducted, using the angular distance information between the candidate navigation star and the calibration star, as well as the reference star map, to determine the recognition results. Experiments demonstrate that when the position noise is 2 pixels, the recognition rate of the algorithm proposed in this paper remains at 96.3%, which is 2.2% higher than that of the optimized grid algorithm [26] and 3.9% higher than that of the regular grid algorithm. Similarly, when the magnitude noise is 0.3 mag, the star identification success rate of the algorithm in this paper remains at 96.2%, surpassing that of the optimized grid algorithm by 1.7% and 3.6%. Therefore, compared to the grid algorithm and the optimized grid algorithm, the proposed algorithm exhibits strong robustness against position noise and magnitude noise.

In summary, the innovations of this paper are as follows: (1) A criterion for selecting multiple calibration stars in grid pattern feature construction is designed, improving the success rate of star selection under noisy interference conditions and thereby enhancing the algorithm’s robustness. (2) Angular distance matching of nearest-neighbor stars is used to introduce more feature dimensions of effective information based on the grid pattern, providing a more comprehensive and effective description of the distribution characteristics of neighboring stars around the main star. (3) By utilizing the angular distance information between the main star and calibration stars to filter the initial identification results, a strategy based on reference star image is designed to efficiently validate the identification results. (4) The characteristics and solution ideas for high-magnitude star identification problems are analyzed, and a solution strategy for identifying high-magnitude star maps within the entire celestial sphere is proposed.

The organizational structure of this article is as follows: The second part introduces the construction of a hybrid grid pattern and presents ideas for algorithm improvement. The third part outlines the specific content of the star recognition algorithm, including initial matching based on mixed grid patterns and the verification of reference stars. The fourth part examines the testing of the algorithm and explains the results of simulation experiments. Finally, the fifth part provides relevant conclusions.

## 2. Algorithm Description

In this section, the algorithm procedure of traditional grid pattern algorithm is first presented, and then its shortcomings in grid pattern construction and pattern matching are also analyzed in detail. Subsequently, some pattern design criteria are given based on above analysis, and then optimization strategies are proposed to refine the pattern design and pattern match procedure to deal with these defects. Lastly, a detailed description of the hybrid grid pattern algorithm, which includes the construction of the hybrid grid pattern, navigation database, and the algorithm’s implementation, is presented.

### 2.1. Introduction to Traditional Grid Pattern Algorithm and Analysis of Limitations

The instruction threshold of the scientific camera is more than 12 mag. This is much higher than that of the traditional star sensor, which is between 5.5 and 6.5 mag. This means that the number of stars that need to be identified by the algorithm for the camera is about more than 100 times the number of stars that need to be identified by a star sensor. Therefore, the technical difficulty in star identification for scientific cameras is far greater than that in star sensor projects. The designed match feature space must have the capacity to contain $2.8\times 10^{6}$ feature vectors of navigation stars. It is also necessary to ensure that the distance between feature vectors in the space is as large as possible to guarantee that they can be distinguished and to make them robust to noise in the feature matching process. In this paper, feature density is introduced to measure the density of feature vectors in the feature space, which is calculated by the number of navigation stars divided by the dimensions of the feature vectors. For subgraph isomorphism algorithms, such as the triangle algorithm or pyramid algorithm, their dimensions of the feature vectors are 3 and 6, respectively. These values are too low to have a small feature density, which means that the probability of redundant or incorrect match is high and that the correct matching result cannot be obtained effectively. In contrast, pattern recognition algorithms in star identification construct a unique pattern based on neighboring stars distributions and their feature dimensions are much higher than those of subgraph isomorphism algorithms. For example, the feature vector dimension in the grid algorithm is set as 1600. Therefore, this technological solution has the potential to achieve feature distinction among navigation stars and is more suitable for star identification tasks with small FOV and high instrument thresholds. The grid pattern exhibits advantages among star identification algorithms based on pattern recognition due to its low distribution density. As a result, the grid pattern is selected as the foundation of star identification for scientific cameras.

The steps of the grid pattern construction for a specific star in are illustrated in Figure 1, and they can be described as follows: (a) select a star to be identified in the captured star image as the primary star, and then determine stars that fall within an annular region defined by circles with radii *r* and *pr* around the primary star; (b) perform shifting on the star image until the primary star is located at the image center, and define the closest neighboring star to the center as the calibration star; (c) rotate the star image so that its horizontal axis is aligned with the straight line that passes through the central primary star and its calibration star; and (d) construct a grid pattern on the basis of this rule: the cells that contain at least one star are considered to be full cells and labeled as 1, while empty cells are labeled as 0. As a result, the grid star pattern can be represented by a one-dimensional grid pattern feature vector (GPFV) $pat=[{cell}_{1,}{cell}_{2,}{cell}_{3,}\ldots {cell}_{k},\ldots {cell}_{g^{2}}]$, assuming that the star image is divided uniformly into g×g cells.
(1)$$\mathrm{fit}(\mathrm{pa}t_{d},\mathrm{pa}t_{s})=\sum_{k=1}^{g^{2}}\mathrm{pa}t_{d}[k]\&\mathrm{pa}t_{s}[k]$$

The similarity score between ${pat}_{s}$ and ${pat}_{d}$ in the matching procedure is estimated using the *fit* function, as shown in Equation (1), where ${pat}_{s}$ is the grid pattern of the primary star in captured star image, ${pat}_{d}$ is the navigation star pattern in the database, “&” is the bitwise and operation, and $g^{2}$ is the total amount of elements contained in the pattern vector. Then, the navigation star with the highest pattern similar score to the pattern of the center star in the star image is considered to be a candidate match result.

There are some reasons that result in the failure of the star identification algorithm. The specific analysis of this issue as follows:

(1) Incorrect selection of the calibration star during the grid pattern construction process is an issue. Choosing the nearest neighboring star as the calibration star to generate the desired grid pattern feature is a crucial step. Some factors that result in the incorrect selection of the calibration star include edge truncation and noise interference. Edge truncation happens when observed stars are located near the edge of FOV, causing their calibration stars to fall outside the imaging area of the detector, which leads to an incorrect choice of calibration star. Noise interference refers to the presence of star position noise and star magnitude noise in the star image, which introduces inconsistencies between the nearest neighboring star points in the observed star image and the data in the navigation database.

(2) The candidate matching result of a primary star in image may not be unique. In the original algorithm, the navigation star with the highest pattern similar score to the pattern of the center star in the star image is considered to be a candidate match result. However, in some cases, there are several navigation stars with the same highest similar score, leading to an inability to identify the primary star in the pattern-matching procedure, especially in a noise interference situation.

(3) The reliability of the identification result based on grid pattern matching is problematic. Even if the pattern is correctly generated and the candidate matching result is unique, the match result may not be correct. Moreover, only the primary star is identified in the algorithm and the calibration star information is not effectively utilized. So, when estimating attitude, there are at least three primary stars identified correctly in the star image, which increases the possibility of star image identification failing.

Currently, several improvement strategies for grid algorithms are available to enhance the success rate of star identification. However, these algorithms still adopt the rule of selecting the nearest neighboring star as the calibration star during the grid pattern construction. As a result, these algorithms based on grid pattern still suffer from the inherent limitation of low accuracy in calibration star selection. Furthermore, none of them fully utilize the information provided by the calibration star, constraining the performance of the algorithm.

### 2.2. Optimization Strategies Based on Traditional Grid Pattern Algorithm

According to the analysis described in Section 2.1, it can be found that the pattern-matching feature selected is a critical part of pattern-based star identification algorithms, greatly influencing the performance of the algorithm. As a result, the ideal design of pattern features should aim to fulfill the following requirements. (1) Reliability: to improve the stability of the observation pattern feature construction process, it is recommended not to select calibration stars in order to avoid generating incorrect patterns based on the calibration star during the process of pattern construction. (2) Uniqueness: the pattern features designed should have ability to describe the distribution relationship between neighborhood stars and primary star sufficiently, namely, the pattern features of different navigation stars should be distinguishable, reducing the probability of incorrect matching or redundant matching. (3) Robustness: the designed pattern should retain the feature elements, even under various noise conditions, ensuring a high identification rate of the algorithm in extreme environments. (4) Simplicity: the pattern construction and matching processes are carried on embedded hardware platform, and so the pattern designed should not be too complex to require a large amount of computational power. In order to optimize algorithm performance and make it more suitable for the star identification scenarios of scientific cameras, we proposed the following three optimization strategies.

(1)Multi-calibration star selection: experimental data from reference [27] indicate that, in real star images, merely 9.6% of primary stars have two nearest neighboring stars that are inconsistent with those of the navigation star database due to noise interference. This is much lower than the inconsistency with the nearest neighboring star. In order to improve the reliability and robustness in pattern construction, the two nearest neighboring stars to the primary star in the captured star image are chosen as the calibration stars, as shown in Figure 2. Then, pattern matching is successively performed based on these calibration stars.(2)The introduction of the neighboring star distance feature: in order to comprehensively characterize the distribution relationship of neighboring stars and increase separability between different patterns, the several brightest neighboring stars falling within the pattern radius of the primary star are selected, and the angular distances between them are recorded into the neighbor star distance feature vector (NSDFV), which supplements the relative distribution features of each neighboring star.(3)Calibration star filter and reference star image verification: to effectively distinguish candidate matching stars with the same similarity score, the distance between the primary star and calibration star is used as a filter to pick the true matched navigation star from among candidate stars. Moreover, the reference star image is generated using identified primary star and its calibration star. Then, it is necessary to verify the identification result by checking the consistency of star positions between the reference star image and the captured star image, which improves the reliability of pattern-matching results.

### 2.3. Hybrid Grid Pattern Algorithm with Multiple Calibration Star Verification

Figure 3 shows the basic process framework of the proposed star identification algorithm, which consists of two major steps: these include an initial match step based on hybrid grid pattern matching and a verification step based on a reference star image. In the initial match step, a hybrid grid pattern feature is constructed by combining GPFV and NSDFV; then, candidate navigation stars are determined by estimating the similarity score between the patterns of the captured star images and databases, which reduces the range of candidate navigation stars. Then, it is necessary to confirm the initial identification result after checking the distance between the primary star and calibration star. In the verification step, a reference star image is generated based on the identification result of the primary star and the calibration stars in the initial match step. Using the principle of nearest-neighbor matching, the star identification task is finished if there are several star pairs meeting the requirements of the nearest star matching principle, which is described in detail in Section 3.

#### 2.3.1. Construction of Hybrid Grid Patterns

The construction of a hybrid grid pattern primarily involves two components: GPFV and NSDFV. By constructing and concatenating these two types of feature vectors, the hybrid grid pattern can be formed. The specific process of constructing the pattern vectors in the navigation database is as follows:

Step 1: take the navigation star at the FOV center as the primary star and determine neighboring stars falling within an annular region defined by circles with radii *r* and *pr* around the primary star.

Step 2: select the closest star to the primary star from the neighboring stars as the calibration star to the construct grid pattern and store the angular distance $d_{pc}$ between the navigation star and calibration star.

Step 3: rotate the star image until the base line connecting the navigation star and calibration star is in a horizontal direction.

Step 4: divide the rotated star image into g×g grids (g = 80). Assign a value of 1 to the grid containing at least one star within its area; otherwise, assign 0 to the grid which does not contain any star within its area. Convert the g×g two-dimensional grid pattern into a vector of length $g^{2}$.

Step 5: calculate the angular distance data among the 15 brightest neighboring stars and then discretize these angular distances. Assign the value of the index corresponding to the discretized distance to 1 in the NSDFV. As shown in Figure 4, the discretized distances between the stars are 61, 96, 98, 102, 105, and 112. As such, the 61th, 96th, 98th, 102th, 105th, and 112th elements in the neighbor star angle distance vector are set to 1. In this paper, the length of NSDFV is set to 200, namely, the discretized distances unit is set to 0.01°.

Step 6: concatenate GPFV and NSDFV to complete the construction of the hybrid grid pattern of the navigation star, as shown in Figure 5.

#### 2.3.2. Navigation Database Construction

The navigation database is the foundation of the algorithm used to identify stars in the entire celestial sphere, and it mainly includes two parts, namely, the navigation star table and the pattern database. The navigation star table contains the star index, magnitude, right ascension, and declination. The pattern database stores the designed hybrid grid pattern feature vector for each navigation star. Moreover, calibration stars’ information, including star index and angular distance to corresponding navigation star, is also stored in the pattern database. The structure of the navigation database is illustrated in Figure 6.

The UCAC4 catalog is a comprehensive and accurate stellar database that provides essential astrometric and photometric information, such as positions, proper motions, and magnitudes, for 113 million stars across the entire celestial sphere, especially for fainter stars whose magnitude is between 8 mag and 15 mag. As such, the UCAC4 is selected as the basic catalog to construct the navigation database. In this study, the limiting magnitude scientific cameras is 12 mag, and so the magnitude threshold-based filtering method is adopted to choose navigation stars, namely, the stars in UCAC4 with magnitude ≤12 are selected as navigation stars, and the corresponding star index, right ascension, and declination data are stored in a corresponding storage location, respectively. After determining the navigation stars in the navigation star table, pattern feature vector construction is performed for each navigation star as described in Section 2.3.1; it should be noted that pattern feature vectors are stored in a lookup table format in the navigation database, as shown in Figure 6. Moreover, during generate grid pattern vectors, the distance $d_{pc}$ between the navigation star and calibration star as well as the star index and position of the calibration stars are all stored in the calibration table.

## 3. Algorithm Implementation

Figure 7 shows the flowchart of the proposed star identification algorithm based on hybrid grid patterns. This includes three primary steps: hybrid grid pattern matching, calibration star filter, and reference star image verification. The first two steps belong to the initial matching stage, while the third step belongs to the validation stage of the framework shown in Figure 3.

In the hybrid grid pattern-matching step, the hybrid grid pattern feature vector is used to determine candidate navigation stars for the captured stars in the image to be identified. However, due to noise interference, there may be redundant or incorrect candidates after hybrid grid pattern matching. To address this issue, a filter method using calibration star information is adopted to remove erroneous results and select the correct result from the candidate navigation stars, which is output as the initial matching result. Then, a reference star image is generated and compared with the captured star image to assess the consistency of star positions. The specific steps of this process are described below.

### 3.1. Hybrid Grid Pattern Matching

Considering the integrity issue of stars at the edge of the image during the pattern construction process, only α (α=15) stars closest to image center are selected as sensor primary stars to be identified using the algorithm. During the pattern match, hybrid grid pattern vectors are constructed and matched for each of the observation stars until one match result meets the requirements for calibration star verification. It should be noted that *n* (*n* = 2) calibration stars are selected and used to generate grid patterns in the captured pattern construction for star images.

Specifically, the nearest-neighbor star to the primary star is selected as the 1st calibration star. Subsequently, the similarity score ${Score}_{p}$ between the hybrid grid pattern generated using the 1st calibration star and the hybrid grid pattern stored in navigation database is calculated as illustrated in Figure 8, where ${Score}_{p}$ is the similarity score between the observation pattern vector in the star image and the pattern vectors in the navigation database and it can be calculated using fast shift and logical AND operation. The navigation stars whose similarity scores are higher than those of the pattern-matching threshold ${Th}_{p}$ are considered to be candidates. If no navigation stars have similarity scores higher than ${Th}_{p}$, then the 2nd nearest-neighbor star is selected as the 2nd calibration star and the abovementioned process is repeated.

After determining candidates, the angular distance feature value between the primary star and calibration star are checked. If the angular distance $d_{pc}^{c}$ between the candidate and its calibration star in the database is close to the angular distance $d_{pc}^{s}$ between the primary star and its calibration star, namely, their error is less than the threshold ${Th}_{c}$, as shown in Equation (2), then the candidates of the primary star and calibration star are determined because initial match results of the algorithm.
(2)$$|d_{\mathrm{pc}}^{s}{-d}_{\mathrm{pc}}^{c}|\leq Th_{c}$$

### 3.2. Reference Star Image Verification

The camera attitude can be estimated using the TRIAD [28] algorithm once the initial match result is determined. Then, a simulated star image under attitude can be generated by combining some camera parameters and the star catalog. It is defined as a reference star image. These only offer star position coordination to accelerate verification step. As shown in Figure 9, red stars represent the locations of stars in the captured star image, and blue stars represent the locations of stars in the reference star image. If the initial match result is correct, then stars’ coordination in the reference star image is close to the corresponding coordination of stars in the captured star image (for example, blue circle indicates a matched pair of stars between the captured and reference star image), and matched star pairs are output as the final identification results of the algorithm. Otherwise, the initial match result is incorrect.

## 4. Experimental Analysis

In this section, the performance of the proposed algorithm is evaluated through experiments on simulation star images and real star images. Furthermore, the test results are compared with those of the grid algorithm and the optimized grid algorithm, both of which perform excellently in star identification and share certain similarities with our algorithm.

In the experiments, the FOV of scientific camera is 2°×2°, the detector resolution is 1024×1024, the pixel size is 10 μm × 10 μm, the focal length is 381 mm, and the detection of limiting magnitudes is set to be 12 mag. Codes about three algorithms are all implemented in MATLAB and run in the PC with a 2.71 GHz Intel Core i7 processor and 16 GB RAM.

The criteria for the successful recognition of a star map are defined as follows: (1) the number of correctly matched observation stars should be no less than 3 and (2) there should be no incorrect matches in the star identification results.

### 4.1. Simulation Experiments

This section primarily focuses on discussing the influence of star position and magnitude noise on the identification rate of different algorithms. In the simulations, one noise source was kept at a typical level while the other was varied linearly. Key parameters remained constant in the experiments. Moreover, the parameters *g* and α were set at the same value for all algorithms. To evaluate the performance of each algorithm, 20,000 simulation star images were captured for each condition using the Monte Carlo method. In addition, the memory size and running time of star identification algorithms are also discussed in this section.

Positional noise is caused due to the optical system’s defects and the sub-pixel centroid algorithm error, making the star centroid deviate from its ideal coordination. In the simulations, star position noise is incorporated into the simulated star images by adding random Gaussian noise to the ideal centroids of the imaged stars. Additionally, the random Gaussian noise is also added to the star brightness to simulate star magnitude noise, which leads to some brighter stars whose magnitude value falls below the instrumental threshold becoming lost, while dimmer stars whose magnitude value is higher than the instrumental threshold actually appear in star images.

#### 4.1.1. Influence of Positional Noise on Recognition Rate

Figure 10 illustrates the variation in identification rates for the multi-calibration star verification hybrid grid pattern algorithm, the optimized grid-based algorithm, and the grid-based algorithm as the standard deviation of positional noise increases from 0.5 pixels to 2.0 pixels. It indicates that the hybrid grid pattern algorithm based on multi-calibration star verification has stronger robustness against positional noise interference in star identification than others. Particularly when the positional noise is large, the algorithm outperforms the original grid-based algorithm. As the standard deviation of positional noise increases from 0.5 pixels to 2.0 pixels, the identification rate of the proposed algorithm remains at 96.43%, whereas those of the optimized grid-based algorithm and the grid-based algorithm are 94.22% and 92.38%, respectively.

The original grid-based algorithm and the optimized grid-based algorithm are more susceptible to positional noise because positional noise can lead to errors in selecting the nearest star as a calibration star during grid pattern construction. In addition, some redundant and incorrect matching happens when using the navigation stars with the maximum similar score. In contrast, the proposed hybrid grid pattern algorithm with multi-calibration reduces the risk introduced by relying on only one calibration star selection through using two nearest-neighbor stars in the pattern construction process. Additionally, the angular distances of the neighboring stars in the pattern feature are independent of calibration star selection. It is necessary to comprehensively describe the distribution feature of neighboring stars and increase the separability among patterns, which enhances the reliability of the pattern matching.

#### 4.1.2. Influence of Magnitude Noise on Recognition Rate

Figure 11 illustrates the variation in identification rates for the multi-calibration star verification hybrid grid pattern algorithm, the optimized grid-based algorithm, and the grid-based algorithm as the standard deviation of magnitude noise increases from 0.1 mag to 0.3 mag. It indicates that the hybrid grid pattern algorithm based on multi-calibration star verification has stronger robustness against magnitude noise, although the identification performance degrades when the added noise standard deviation increases. Specifically, the proposed algorithm’s identification rate declines from 98.85% to 96.45%, and the optimized grid-based algorithm’s identification rate decreases from 97.57% to 94.42%, and the grid-based algorithm’s identification rate drops from 96.82% to 92.63%.

There are several reasons causing the proposed algorithm to perform better with the magnitude of noise interference: (1) The magnitude of noise reduces the accuracy rate of selecting the calibration star in the pattern construction of the original and optimized grid pattern, generating erroneous grid pattern features which cannot be identified correctly. In addition, there have to be at least three correct candidates after the initial match to make sure the identified results are correct. (2) The proposed algorithm enhances the reliability of the identification results by adding calibration star verification after pattern matching. Moreover, the proposed algorithm utilizes reference star images to verify initial matching results after determining attitude, which means only one correctly matched primary star is enough to complete the star identification task. Therefore, the proposed star identification algorithm exhibits stronger robustness against magnitude noise and is more suitable for star identification by scientific cameras with higher apparent magnitude limits.

#### 4.1.3. Run Time and Memory Size

The performance results of our method and the other two algorithms in terms of memory size and running time are presented in Table 1. The results show that the run-time of the proposed method is 2.1 s, which is much faster than that of the grid method (6.3 s) and optimized grid method (4.7 s). This is principally because the proposed algorithm only needs one correctly matched result in identifying α primary stars to consequently complete the star identification task, whereas the other two algorithms have to identify all α primary stars.

The memory maintained by the algorithm refers to the on-board navigation star table and the pattern database, which account for the majority of the memory required for algorithm operation, as described in Section 2.3.2. For each entry in the on-board catalog, four bytes of storage are allocated to store the right ascension and declination values of the navigation star. Therefore, the total memory required for the on-board navigation catalog can be calculated as 12 N, where N represents the number of entries in the catalog. Each entry included in the on-board pattern features consists of navigation star indexes which contain element value 1 in location of this grid. Assuming that the average neighboring stars number in a grid pattern corresponding to an on-board catalog entry is a, then the total number of indexes stored in the lookup table about grid pattern will be Na. Similarly, assuming that the average number of angular distances recorded in the feature vectors of neighboring stars for each navigation star is b, the total number of indexes stored in the lookup table about angular distances among neighboring stars will be Nb. Given that an index is stored using 4 bytes, then the memory size required for the lookup table can be estimated as 4 N(a + b). For the proposed algorithm, N = $2.8\times 10^{6}$, a = 200, and b = 105. Then the total memory size of navigation database for the algorithm is 12 N + 4 N(a + b), which is about 3.5 Gbytes approximately, while that of the grid algorithm and optimized grid algorithm is about 2.3 Gbytes and 1.7 Gbytes because only the grid pattern is stored in the database.

### 4.2. Real Star Image Testing

In addition to the simulation test presented in Section 3.2, the proposed algorithm is also implemented on real star images captured by the scientific camera. Its FOV is 2°×2° and its limiting magnitude detection is 12 mag. The star centroiding method is first utilized to estimate stars’ coordinates in images, and then the star identification algorithm is performed on the 98 real star images captured using different camera boresight orientations.

The experiment result indicates that the proposed algorithm successfully identified all real star images.

Examples of real star image identification are presented in Figure 12, in which the stars’ positions in images, estimated via the centroiding method, are marked with red circles, while green circles show the matched reference stars’ positions in the star image. It can be seen that the positions of the reference stars almost overlap with the positions of the stars in the captured star image, and so the star image is identified successfully. Moreover, the red numbers outside the brackets and inside the brackets are the star index and the stellar magnitudes in the UCAC4 catalog.

## 5. Conclusions

This paper proposes a full-sky autonomous star identification algorithm based on a hybrid grid pattern for scientific cameras. Several optimization strategies are designed to achieve star identification with the characteristics of a 2° small field of view and a 12.0 mag high limiting magnitude in the algorithm. The multi-calibration stars are selected to improve the model’s reliability and robustness in pattern construction, and the distance feature for neighboring stars is also introduced to comprehensively characterize the distribution relationships of neighboring stars and increase the separability between different patterns. Moreover, a calibration star filter is adopted to distinguish candidate matching stars in the initial match, and then the reference star image is generated to verify and determine the final identification results, which greatly enhances the reliability of the algorithm in dealing with noise. Simulation experiments demonstrate that the proposed algorithm exhibits superior robustness against positional and magnitude noise when compared to the optimized grid algorithm and the grid algorithm. Additionally, the proposed algorithm achieves a higher identification rate on tested real star images in comparison to the optimized grid-based algorithm. With its outstanding ability to identify dim stars within a small field of view, the proposed algorithm shows great potential for use in various space-related applications. In the future, we will focus on applying this technology to areas such as telescope precision navigation systems, space debris detection, and space-based digitized sky surveys.

## Figures and Tables

**Figure 1 sensors-24-01661-f001:**
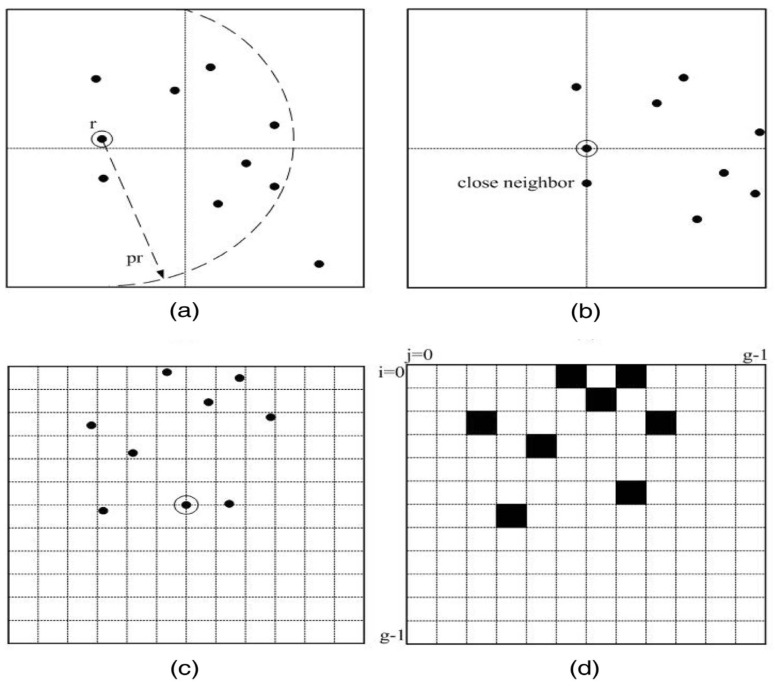
The traditional grid pattern construction. (**a**) select a star to be identified in the captured star image as the primary star; (**b**) perform shifting on the star image until the primary star is located at the image center; (**c**) rotate the star image, then build the grid; (**d**) Complete star point selection based on established rules.

**Figure 2 sensors-24-01661-f002:**
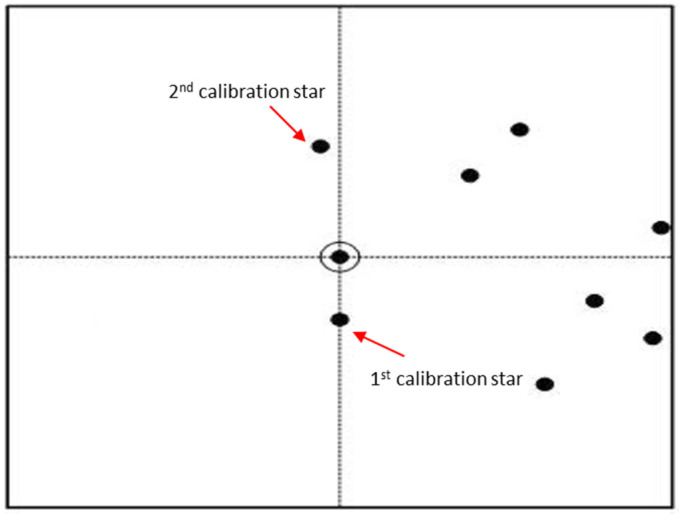
The schematic diagram of selecting multi-calibration stars.

**Figure 3 sensors-24-01661-f003:**
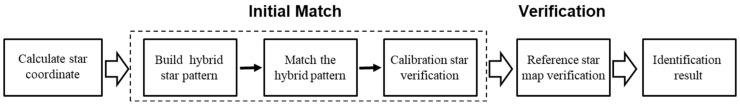
The algorithm framework.

**Figure 4 sensors-24-01661-f004:**
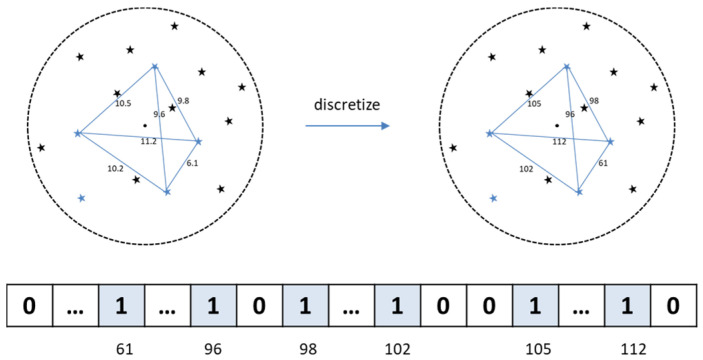
Construction of neighbor star angular distance vector.

**Figure 5 sensors-24-01661-f005:**
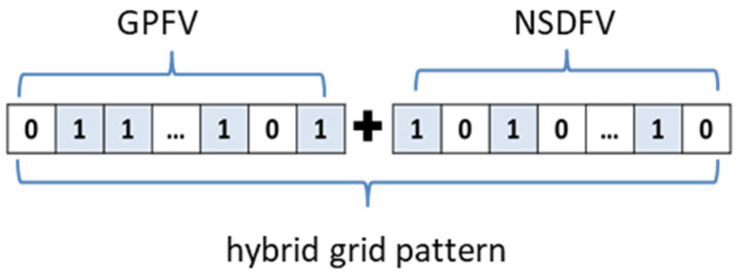
The schematic diagram of hybrid grid mode splicing.

**Figure 6 sensors-24-01661-f006:**
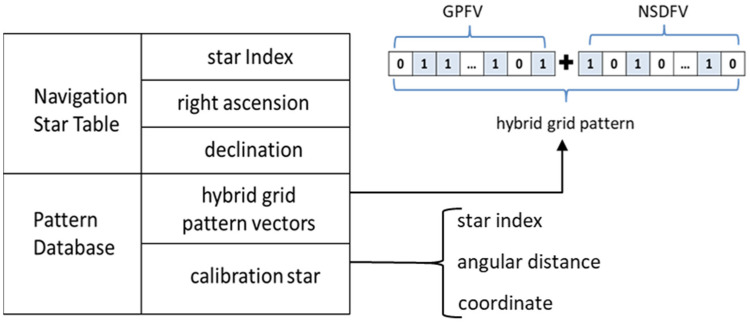
The feature pattern database structure.

**Figure 7 sensors-24-01661-f007:**
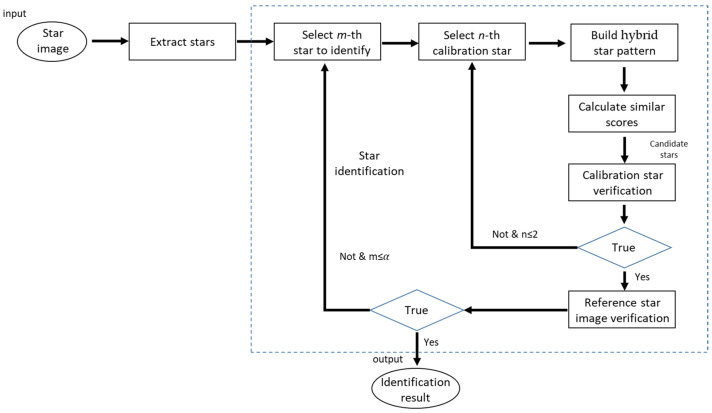
Flowchart of our algorithm.

**Figure 8 sensors-24-01661-f008:**
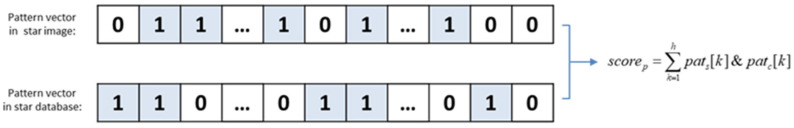
The schematic diagram of pattern feature vector similarity calculation.

**Figure 9 sensors-24-01661-f009:**
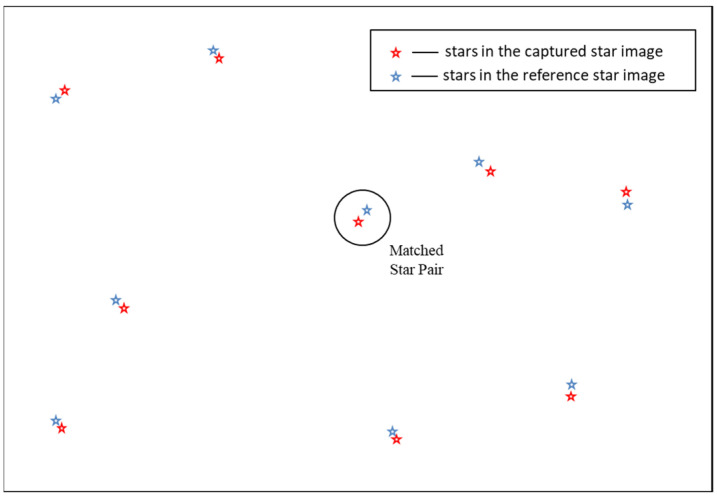
The schematic diagram of reference star image verification.

**Figure 10 sensors-24-01661-f010:**
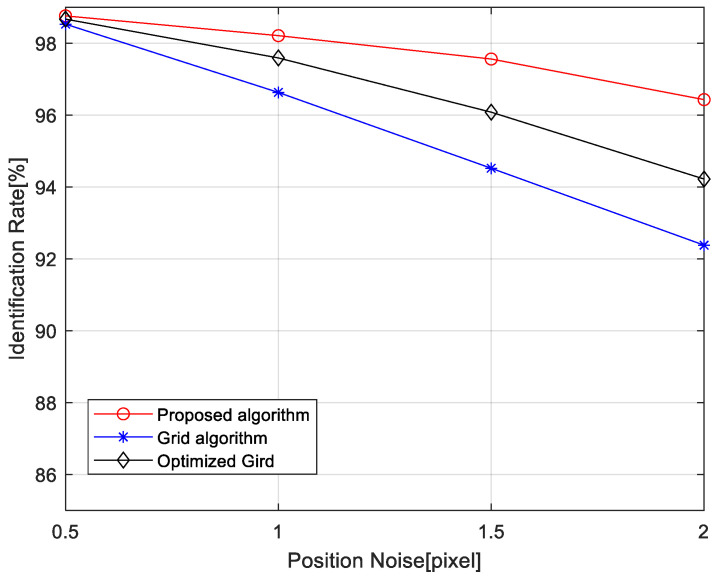
Identification rate versus positional noise.

**Figure 11 sensors-24-01661-f011:**
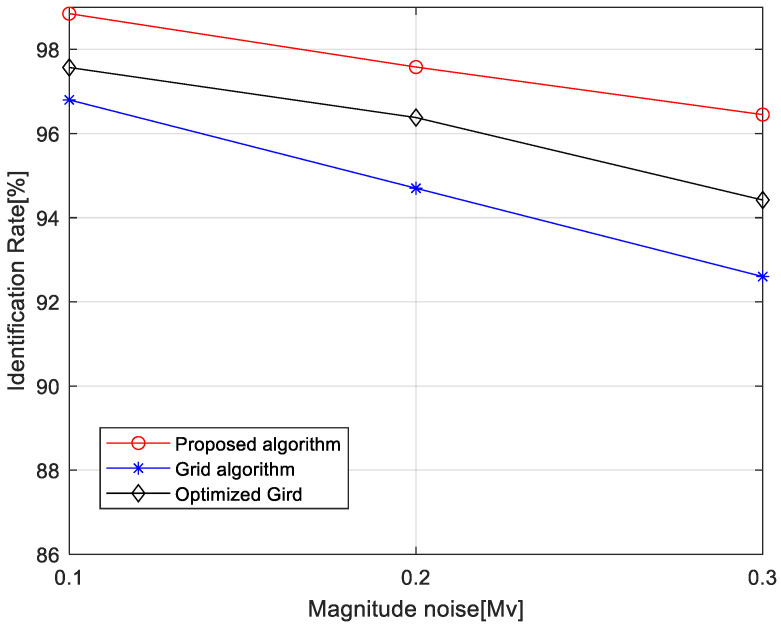
Identification rate versus magnitude noise.

**Figure 12 sensors-24-01661-f012:**
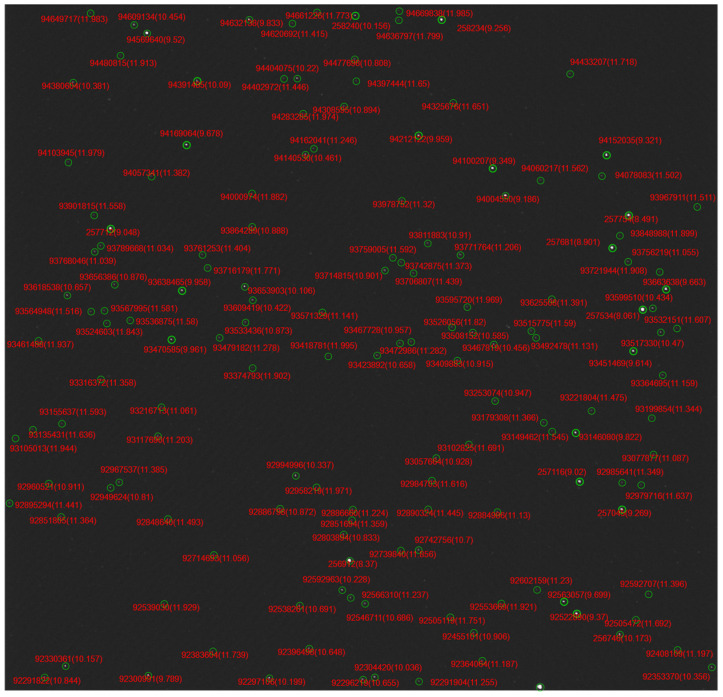
An example of identification results in a real star image.

**Table 1 sensors-24-01661-t001:** Time and Memory Size.

Algorithm	Our Method	Optimized Grid	Grid Method
Run time	2.1 s	4.7 s	6.3 s
Memory cost	3.5 GB	2.3 GB	1.7 GB

## Data Availability

Data are contained within the article.

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
