# Peer review of "Hybrid Grid Pattern Star Identification Algorithm Based on Multi-Calibration Star Verification"

_sensors, 2024, doi:10.3390/s24051661_

Round 1

Reviewer 1 Report

Comments and Suggestions for Authors

The authors provide a new algorithm for a star sensor, which resolves a star field and estimates the attitude of a spaceship. The referee highly recognizes the advantage and applicability of the proposed algorithm. However, the description of the proposed algorithm should be updated so that readers can implement it. The experimental designs were not well described, and the results of the experiments were poorly presented. The size of the navigation star database should be discussed in comparison with some subgraph isomorphic methods. The referee supposes that these major issues should be properly addressed before the publication.

Comments on the Quality of English Language

The manuscript contains a number of grammatical errors: wrong choices of words, tense, articles, and subject-verb agreements. There are many sentences that do not make sense. Such poor descriptions deteriorate the quality of the manuscript. The current version is far from acceptable and hard to review. The referee strongly recommends that the authors check every sentence before re-submission. Otherwise, the authors should consider using a proofreading service.

Author Response

Thanks very much for your comment, I have written my response in the attachment below

Reviewer 2 Report

Comments and Suggestions for Authors

This manuscript introduces an innovative star identification algorithm utilizing a hybrid grid pattern and multi-calibration star verification. The algorithm forms a mixed grid pattern feature vector, combining a grid pattern vector with a neighbor star angle distance vector to depict the star distribution around a primary star. It incorporates multiple calibration stars in the pattern construction to boost robustness. The algorithm initially matches patterns to identify candidate stars, which are then verified through calibration star angular distance. A reference star map created from these results further validates the recognition process. The algorithm demonstrates notable robustness against position and magnitude noise compared to existing grid algorithms in simulation experiments.

I recommend the following enhancements for the manuscript:

1. Introduction Revision: Expand the introduction to include a comprehensive review of existing star identification algorithms. This should highlight the motivation and distinctiveness of your proposed approach, with a detailed discussion on the shortcomings of current methods.

2. Algorithm Workflow Clarification: In Figure 3's description of the algorithm workflow, clearly indicate why the algorithm selects the "a" closest stars to the center for initial recognition, addressing potential edge incompleteness. Include specific values for "a" tested in your study.

3. Criteria for Navigation Star Selection: Define precise criteria for selecting navigation stars from the UCAC catalog. The current description of choosing stars with "apparent magnitude parameters less than or equal to 12" is somewhat ambiguous.

4. Experimental Methodology Enhancement: Improve the experimental section by detailing the PC specifications and software used for simulations. Consider increasing the sample size for statistical testing to bolster the significance of your recognition rate findings. Additionally, evaluate and report on the computational complexity of the algorithm.

5. In-depth Analysis of Results: Expand the analysis of experimental outcomes beyond mere recognition rates. Delve into the factors influencing performance and conduct a failure case analysis to uncover any limitations.

6. Conclusions and Real-world Applications: In the conclusions, focus on summarizing the key empirical advantages of your approach. Discuss potential applications in real-world scenarios to underscore the practical significance of your findings.

7. Manuscript Proofreading: Thoroughly proofread the manuscript to correct any typos, formatting inconsistencies, and citation errors. Ensure that the manuscript complies with the author guidelines of your target journal.

Comments on the Quality of English Language

Editing of English language required

Author Response

Thanks very much for these comment, I have written my response in the attachment below
